# The effect of cyberbullying on nonsuicidal self-injury in adolescents: The chain mediating role of shame and dissociation

Bin Huang[1], Jixu Yin[1], Qianmei Long[2], Jia Yu[1], Junlin Wu[3], Guoping Huang[3]*

1 Department of Psychosomatic Diseases, The Third Hospital of Panzhihua, Panzhihua Mental Health Center, Panzhihua, China, 2 Intensive Psychiatric Unit,The Fourth People's Hospital of Chengdu, Chengdu Mental Health Center, Chengdu, China, 3 Editorial Department, The Third Hospital of Mianyang, Sichuan Mental Health Center, Mianyang, China

☯ These authors contributed equally to this work and should be considered co-first authors
* cahuanggp@163.com

## Abstract

Adolescents are prone to nonsuicidal self-injury, a unique risk factor for suicide and suicide attempts. Prior research has shown that cyberbullying predicts adolescent nonsuicidal self-injury behavior. Therefore, this study aimed to develop a chain mediation model to investigate the impact of shame and dissociation related to cyberbullying on teenage nonsuicidal self-injury. Between 23/04/2022 and 26/04/2022, researchers recruited 14,666 high school students in Zizhong County, Sichuan Province, China, to investigate cyberbullying, non-suicidal self-injurious behaviors, shame, and feelings of dissociation through self-report questionnaires. Among the participating high school students, 56.0% reported experiencing cyberbullying and 25.4% reported nonsuicidal self-injurious behaviors. After adjusting for the effects of gender, traditional bullying, grade level, left-behind child status, and parental marital status, there was a positive correlation between cyberbullying and adolescent nonsuicidal self-injurious behavior. Shame and dissociation played a mediating role between cyberbullying and adolescent nonsuicidal self-injury. First, a partial mediating effect of shame and dissociation was observed, with the mediating effect accounting for 6.1% and 21.2% of the total effect (33.0%), respectively; thereafter, a chain mediating effect of shame and dissociation was noted, with the mediating effect accounting for 9.1% of the total effect. In the parallel mediation test, the mediating effect of dissociative experience (0.10) was higher than that of shame (0.02). Cyberbullying and non-suicidal self-injury are prevalent among high school students in western China, and the synergistic effects of shame and dissociation may be associated with increased risk of adolescent non-suicidal self-injury.

**Data availability statement:** All relevant data are within the manuscript and its Supporting Information files.

**Funding:** This work was supported by the Panzhihua City Guiding Science and Technology Program: Application and Efficacy Observation of Meaning Therapy in Children and Adolescents with Depression (2022ZD-S-51). The funders had no role in study design, data collection and analysis, decision to publish, or preparation of the manuscript.

**Competing interests:** The authors have declared that no competing interests exist.

## Introduction

### Cyberbullying and nonsuicidal self-injury

Cyberbullying is defined as an act in which an individual or group repeatedly or intentionally disseminates offensive words or images through electronic media, including email, instant messaging, bulletin board system forums, chat rooms, mobile devices, social networking sites, weblogs, and WeChat groups [1]. Although there is overlap between traditional bullying and cyberbullying—with approximately 85% of youths who engage in cyberbullying also involved in traditional forms [2]—several features distinguish cyberbullying from its traditional counterpart. Cyberbullying is carried out through electronic media, which increases its occurrence due to easy accessibility, wide coverage, and rapid dissemination. It is unrestricted by time and space, can occur anywhere and anytime, and amplifies negative consequences [3]. Moreover, because cyberbullying does not occur face-to-face, perpetrators can attack others anonymously [4,5], and the inability to accurately assess power dynamics renders everyone a potential target [6]. Perpetrators do not witness the emotions of remote victims and may experience less guilt and remorse, leading to repeated or continuous cyberbullying [3].Research indicates that females, members of minority groups, individuals with neurotic tendencies and low agreeableness, as well as those exhibiting higher levels of anxiety, depression, and anger are more likely to become victims of cyberbullying [7–10]. With prevalence rates reaching up to 65% among adolescent victims, cyberbullying is common and associated with several detrimental outcomes, including anxiety, depression, sleep disturbances, substance addiction, and suicide [11–14]. It can also significantly increase a victim's likelihood of engaging in self-harming behavior [15].

Nonsuicidal self-injury (NSSI) refers to the direct, repetitive act of hurting oneself without suicidal intent, which is not socially acceptable. Such acts include cutting, scratching, burning body surfaces, and hitting objects with one's body to cause bleeding, bruising, or pain [16]. Adolescents frequently engage in NSSI; in a global school sample, the lifetime prevalence of at least one NSSI act averaged 17.2% (range: 8.0–26.3%) [17], and 12.25% of adolescents reported repeated NSSI [18]. The overall prevalence of NSSI among Chinese secondary school students was 22.37%, with rates increasing annually [19,20]. Although NSSI is performed without suicidal intent, repeated NSSI directly increases suicide risk and can serve as an independent risk factor for suicide [21,22]. Given the high detection rate of NSSI among Chinese adolescents and its numerous negative consequences, understanding the mechanisms underlying NSSI is of considerable clinical importance. The present study aimed to examine the association between NSSI and cyberbullying in a large sample of high school students and to explore potential psychological mechanisms underlying this relationship.

### The potential mediating role of shame

Shame is a powerful, painful, self-directed emotion arising from negative evaluations of oneself and one's environment, which may target one's personality,

behavior, ability, appearance, or other attributes [23]. Being bullied constitutes a traumatic event [24], and shame represents a typical emotional reaction to trauma [25–27]. Objectification theory provides a valuable framework for understanding the link between appearance-focused cyberbullying victimization and shame. The theory posits that repeated exposure to external scrutiny—such as being mocked online for one's appearance—can lead individuals to internalize an observer's perspective on their own bodies, a process known as self-objectification [28]. In the context of cyberbullying, this heightened self-surveillance and body monitoring can evoke intense shame, as individuals perceive themselves as failing to conform to socially imposed aesthetic standards [29]. For example, individuals subjected to appearance-focused cyberbullying become more concerned about their appearance and develop greater body shame [29,30]. The intrinsic negative reinforcement function of NSSI—reducing or distracting from unpleasant thoughts or sensations—positions shame as a crucial emotion in NSSI [31], as individuals may engage in NSSI to avoid experiencing the painful emotion of shame [32]. Consequently, shame may mediate the relationship between cyberbullying and NSSI.

### The potential mediating role of dissociation

Dissociation is a blunted affective state characterized by disruption and/or interruption of the normal integration of consciousness, memory, identity, emotion, perception, physical performance, motor control, and action [16]. Adolescence represents a critical period for the development of dissociative behaviors and the refinement of fundamental mental functions [33]. Bullying constitutes a traumatic experience that can lead to post-traumatic stress symptoms, including dissociation. Previous studies have identified a connection between the frequency and intensity of dissociation and deliberate self-injury in adolescents [34]. Individuals may engage in NSSI to escape unpleasant emotions by inducing dissociation, or alternatively, they may use NSSI to terminate dissociation when experiencing an unreal or disconnected state [35]. Few studies have examined the association between dissociative episodes and cyberbullying specifically. The present study hypothesizes that dissociative experiences may be linked to cyberbullying and may serve as mediators between cyberbullying and NSSI.

### The potential chain mediating effects of shame and dissocation

Meta-analyses have demonstrated that dissociation is linked to shame [36]. Increased shame is associated with greater dissociative symptoms, and dissociative tendencies predict non-suicidal self-injury during preadolescence and adolescence [37,38]. Accordingly, we anticipated that shame and dissociation would function as chain mediators in the relationship between cyberbullying and NSSI.

### Gender differences

In light of established gender differences—with females showing greater vulnerability to cyberbullying victimization [9] and NSSI [39], and specific susceptibility to shame arising from appearance-related attacks [40]—we treated gender as a covariate. This statistical control was implemented to account for its potential confounding influence, thereby allowing more precise estimation of the specific chain-mediating effects of shame and dissociation, the latter of which shows no marked gender variation [41].

### The current study

Due to the co-occurrence of traditional bullying and cyberbullying, focusing solely on cyberbullying may overstate its negative effects. After controlling for the impact of traditional bullying on NSSI, this study aimed to examine the chain mediating roles of shame and dissociation in the relationship between cyberbullying and adolescent NSSI among secondary school students in Zizhong County, Sichuan Province, China.

## Materials and methods

### Participants

The target population was high school students in Zitong County, Sichuan Province. 14,666 questionnaires were distributed from 23/04/2022–26/04/2022, and 14,036 valid questionnaires were collected, with a recovery rate of 95.7%. Of the participants, 5629 (40.1%) were senior high school students, followed by 5304 (37.8%) sophomores, and 3103 (22.1%) seniors. Males comprised 45.1% of the participants (n = 6326), whereas females comprised 54.9% (n = 7710). Ultimately, 3573 individuals (25.5%) reported NSSI in the previous year, with 58.0% of females and 42.0% of males reporting that they had experienced cyberbullying at least once in the previous three months.

### Procedure

This study was approved by the Ethics Committee of Mianyang Third People's Hospital (Approval Document No:2022 Annual Review [10]). The study obtained passive parental consent and active student informed consent. Prior to data collection, a written study description was sent to parents/guardians, who could opt their children out of the study by returning the form. On the survey day, students received a detailed information sheet explaining the study's purpose, the principle of voluntary participation, anonymity, and the right to withdraw at any time without consequences. All participating students signed a written consent form. The survey was administered by trained researchers in a controlled classroom setting, emphasizing the confidentiality of responses. Questionnaires were completed anonymously and collected separately from any identifying information. Data was stored on a password-protected secure server accessible only to the research team.

### Measures

Self-designed general demographic questionnaire for participants: gender (male, female), age, grade level (10th grade, 11th grade, 12th grade), left-behind status (yes, no), parental marital status (married, divorced, widowed).

cyberbullying was assessed by the Revised Cyber Bullying Inventory [42]. The cyberbullying subscale consists of 14 questions, each measuring the frequency at which the participant experienced cyberbullying behavior over three months. Examples of these experiences include "Someone posted fake photos or messages online to defame me," "Someone stole my personal information from my computer or phone," "Someone abused me online," and "Someone threatened me online." The assessment is based on a 4-point scale as follows: 1, "never encountered," 2, "once," 3, "twice or Three times," and 4, "more than thrice." The more frequently the participants experienced cyberbullying, the higher the overall score. If a participant responded "once," "twice or thrice," or "more than thrice," cyberbullying was deemed to have occurred. According to this study, the updated Chinese version of the questionnaire is valid and reliable and can be used in a Chinese cultural setting. Cronbach's alpha coefficient for this scale in this study was 0.844.

Traditional bullying victimization was assessed by the bullying sub-questionnaire of the Olweus Bully/Victim Questionnaire revised by Wenxin Zhang [43], which includes six topics including "Others give me nasty nicknames," "Certain classmates hit, kick, push, or bump me," and "Some classmates spread rumors about me and try to make others dislike me." The questionnaire included two questions for each of the three types of bullying—physical, verbal, and relational—and asked participants to rate how frequently they had experienced bullying over the previous three months, on a scale of 0–4 (0: never experienced bullying, 1: experienced bullying only once or twice, 2: experienced bullying twice or thrice a month, 3: experienced bullying once a week, and 4: experienced bullying several times a week). A participant was judged to have experienced traditional bullying if their response was anything other than "0." The higher the score, the more severe the amount of traditional bullying they experienced. Cronbach's alpha coefficient for this scale of the questionnaire used in this study was 0.644.

Shame was measured by the Shame Scale for Secondary School Students [23], which was developed by Qi et al. to match the behavioral characteristics of secondary school students. Using four dimensions—personality shame,

behavior shame, body shame, and ability shame—and a total of 22 entries, the scale is used to assess the feelings of shame in secondary school students during the previous year. Questions included "Have you ever felt regrettably ashamed of the kind of person you are, as well as your physical prowess and memory?" and "Have you ever felt humiliated because of your shape or physical abilities?" Shame susceptibility was measured using a 4-point scale, with a higher overall score indicating stronger vulnerability. The Cronbach's alpha coefficient for this scale in this study was 0.98.

Dissociative experiences were assessed using the Adolescent Dissociative Experiences Scale (A-DES [33]), a modification of the DES with proven reliability. The A-DES has 30 questions about dissociative experiences, including "When I am watching TV, reading a book, or playing online games, I am very absorbed and lose track of my surroundings" and "I also experience powerful emotions that don't appear to be a part of me." Each response was given a score between 0 and 10, with 0 denoting "never had this experience" and 10 denoting "had this experience recently." Items were scored by adding their scores, dividing by 30, and averaging the results to obtain a total score out of 10. The degree of dissociation increases as the score increases. The scale used in this investigation has good internal consistency, with a Cronbach's alpha coefficient of 0.945.

NSSI was assessed using the Adolescent Self-Injurious Behavior Questionnaire developed by Zheng Ying and revised by Feng Yu [44]. This questionnaire is a self-assessment questionnaire, measuring a total of 18 items, each of which represents a method of self-harm by the participant, such as "deliberately scratching one's skin with glass or a knife," "deliberately poking open wounds and interfering with wound healing," "intentionally burning/scalding one's skin," and "intentionally hitting oneself with fists or harder objects." Additionally, this survey included two assessment dimensions: the frequency of participants intentionally injuring themselves in this way in the previous year, measured using a 4-point scale with 0 times, one time, two to four times, and more than five times, and the severity of their resultant physical injury, measured using a 5-point scale including none, mild, moderate, severe, and very severe. The product of the number of self-injuries and degree of self-injury over 18 items yielded an assessment index of the level of self-injury. The composite score (frequency × severity) was calculated to reflect the overall burden of NSSI behavior, a method used in prior adolescent NSSI research to capture both behavioral frequency and physical consequence [45]. This approach provides a single index that accounts for both dimensions of self-injurious behavior. The severity of self-injurious behavior increased with the score. Patients had a history of self-injury if their total score was higher than 0, and Cronbach's alpha coefficient for this scale of the questionnaire was 0.840.

Although both the Shame Scale for Secondary School Students and the Adolescent Self-Injurious Behavior Questionnaire demonstrated good reliability, this study was unable to conduct separate analyses of the subscales within each measure due to the lack of detailed operational definitions for these subcomponents and validated factor structures in the existing literature.

## Data analysis

Self-reporting techniques were used to obtain data for this study, which raises the possibility of common method bias. Therefore, Harman's one-way test was used. All questions from each questionnaire were included in the exploratory factor analysis using SPSS 26.0 (IBM SPSS Statistics for Windows, Armonk, NY, USA: IBM Corp.), which revealed that the first factor's cumulative variance explanation was 21.13%. As this is less than 40%, this indicated that there was no significant common method bias. Spearman's Correlation Analysis and Partial Correlations Analysis was used to test correlations between the main variables. Model 6 in SPSS plug-in PROCESS 4.0, developed by Haynes [46], was used to test for chain mediated effects, and Model 4 was used to test for parallel-mediated effects [47]. All data analyses were carried out using SPSS 26.0 software, reported as mean ± standard deviation in descriptive analyses, and two-tailed tests were used for all statistical models.

## Results

### Descriptive statistics and bivariate correlations

As shown in Table 1, male students reported significantly higher scores on the cyberbullying and traditional bullying scales, while female students scored higher on measures of shame, dissociation, and NSSI. In addition to gender, this study collected demographic data on grade level (10th, 11th, and 12th grade), left-behind status (defined as having at least one parent absent for more than six months), and parental marital status (married, divorced, or widowed). Further analyses revealed significant associations between these demographic factors and the key outcome variables. Specifically, one-way ANOVA results indicated statistically significant differences across grade levels in both cyberbullying ($F = 199.01$, $p < 0.05$) and NSSI scores ($F = 166.63$, $p < 0.05$), with scores highest among 10th graders and lowest among 12th graders. Independent t-tests showed that left-behind students scored significantly higher than their non-left-behind peers on both cyberbullying ($t = 72.04$, $p < 0.05$) and NSSI ($t = 3.77$, $p < 0.05$). Regarding parental marital status, students with widowed parents reported the highest levels of cyberbullying, whereas those with divorced parents exhibited significantly higher NSSI scores compared to students from married households.

Of the participants, 2194 (15.6%) reported experiencing cyberbullying at least once in the past three months; 4553 (32.4%) participants reported being both victims of cyberbullying and traditional bullying; 3573 participants (25.5%) reported self-injurious behaviors in the past year, of which 747 (20.9%) experienced cyberbullying, 464 (13.0%) participants who had experienced traditional bullying reported NSSI s, and 1787 (50.0%) participants who had experienced NSSI s reported both cyberbullying and traditional bullying, with a statistically significant difference ($\chi^2 = 132.023$, $p < 0.01$). Splitting participants into groups with and without NSSI revealed that the group with NSSI scored higher than the group without NSSI in terms of cyberbullying, traditional bullying, humiliation, and dissociation. Spearman's correlation analysis revealed correlations between self-harming behaviors, shame, traditional bullying, and cyberbullying. Research findings indicate that after controlling for variables such as gender, traditional bullying, grade level, left-behind child status, and parental marital status, significant correlations persist among the variables. Cyberbullying remains significantly positively correlated with non-suicidal self-injury behaviors.The specifications are listed in Table 2.

### Chain mediated effects test

The analysis, which was based on the bias-corrected percentile bootstrap method, was conducted using the SPSS plug-in PROCESS 4.0 [46]. Each variable was first standardized, and the 95% confidence interval was calculated by repeating the sample 5000 times while controlling for gender, traditional bullying victimization, grade level, left-behind child status, and parental marital status. The regression analysis results are presented in Table 3. Four models were created to examine the impacts of cyberbullying, shame, and dissociation on NSSI after controlling for the effects of gender and traditional bullying victimization. Model 1 showed that cyberbullying was significantly positively associated with NSSI

**Table 1. Differences in the main variables by gender and by NSSI status (N = 14,036).**

| Variable | M ± SD | | Z | p | Cohen's d | M ± SD | | Z | p |
|---|---|---|---|---|---|---|---|---|---|
| | Male (n = 6326) | Female (n = 7710) | | | | NSSI (n = 3573) | No NSSI (n = 10,463) | | |
| Cyberbullying score | 16.68 ± 4.22 | 16.02 ± 3.42 | 10.27 | <.01 | 3.80 | 17.90 ± 5.03 | 15.78 ± 3.12 | 27.75 | <.01 |
| Traditional bullying score | 7.23 ± 2.04 | 6.80 ± 1.51 | 16.56 | <.01 | 1.77 | 7.68 ± 2.39 | 6.76 ± 1.44 | 27.65 | <.01 |
| Shame score | 48.53 ± 12.57 | 50.02 ± 13.11 | 15.55 | <.01 | 12.87 | 57.74 ± 12.31 | 47.96 ± 12.25 | 38.32 | <.01 |
| Dissociation score | 2.07 ± 1.71 | 2.10 ± 1.75 | 0.49 | 0.62 | 1.74 | 3.19 ± 2.00 | 1.70 ± 1.46 | 43.01 | <.01 |
| NSSI score | 1.50 ± 5.23 | 2.09 ± 6.64 | 5.14 | <.01 | 6.04 | | | | |

NSSI, nonsuicidal self-injury; M, Mean number; SD, Standard Deviation

**Table 2. Descriptive statistics and bivariate correlations of main variables (N = 14,036).**

|  | Variable | M | SD | 1 | 2 | 3 | 4 |
|---|---|---|---|---|---|---|---|
| 1 | Cyberbullying | 16.32 | 3.82 | 1.00 |  |  |  |
| 2 | Shame | 50.45 | 12.99 | 0.17** | 1.00 |  |  |
| 3 | Dissociation | 2.08 | 1.74 | 0.23** | 0.46 ** | 1.00 |  |
| 4 | NSSI | 1.82 | 6.05 | 0.19** | 0.20** | 0.31** | 1.00 |

*$p<0.05$, **$p<0.01$; NSSI, nonsuicidal self-injury; M, Mean number; SD, Standard Deviation

**Table 3. An analysis of the mediating role of shame and dissociation in the relationship between cyberbullying and NSSI (N = 14,036).**

| Model | Outcome variable | Predictor variable | R | R² | F | β | t | Boot LLCI | Boot ULCI |
|---|---|---|---|---|---|---|---|---|---|
| Model 1 | NSSI | Cyberbullyin | 0.37 | 0.14 | 370.88*** | 0.20 | 22.69*** | 0.184 | 0.219 |
| Model 2 | Shame | Cyberbullyin | 0.35 | 0.12 | 323.98*** | 2.41 | 20.68 *** | 2.183 | 2.640 |
| Model 3 | Dissociation | Cyberbullyin | 0.56 | 0.32 | 929.47*** | 0.17 | 20.58*** | 0.150 | 0.181 |
|  |  | Shame |  |  |  | 0.33 | 57.20*** | 0.032 | 0.034 |
| Model 4 | NSSI | Cyberbullyin | 0.45 | 0.21 | 452.37 *** | 0.13 | 14.66 *** | 0.112 | 0.146 |
|  |  | Shame |  |  |  | 0.004 | 6.97 *** | 0.003 | 0.006 |
|  |  | Dissociation |  |  |  | 0.25 | 27.756*** | 0.233 | 0.269 |

*$p<0.05$, **$p<0.01$, ***$p<0.001$;NSSI, nonsuicidal self-injury; Boot LLCI, lower limit of confidence interval; Boot ULCI, upper limit of confidence interval;All models controlled for gender, traditional bullying victimization, grade, left-behind status, and parental marital status.

($\beta=0.20, p<0.001$), and it explained 14.0% of the variance in adolescent NSSI. In models 2 and 3, cyberbullying was positively associated with shame ($\beta=2.41$, $p<0.001$) and dissociation ($\beta=0.17$, $p<0.001$), respectively; model 2 showed that shame was positively associated with dissociation ($\beta=0.33$, $p<0.001$). In Model 4, shame and dissociation were significantly positively associated with adolescent NSSI ($\beta=0.004$, $p<0.001$ and $\beta=0.25$, $p<0.001$, respectively).

The mediation analysis results indicate that the total indirect effect of shame and dissociation in the relationship between cyberbullying and NSSI was 0.07, explaining 35.0% of the total effect (0.20 and 53.8% of the direct effect (0.13).The mediating effect consisted of the indirect effects generated by the following three paths: Ind1: cyberbullying→shame→self-injurious behavior; Ind2: cyberbullying→dissociation→self-injurious behavior; Ind3: cyberbullying→shame→dissociation→self-injurious behavior; the confidence intervals of all three paths did not contain 0 values, indicating that the indirect effects generated by all three paths reached the significance level. The results are presented in Table 4 and Fig 1.

In addition, after controlling for the effects of gender, traditional bully ingvictimization, grade level, left-behind child status, and parental marital status, we constructed three equations to examine and compare the mediating role of shame and dissociation in cyberbullying and NSSI. Table 5 shows that in Models 1 and 2, cyberbullying was significantly and positively associated with shame ($\beta=2.41$ $p<0.001$) and dissociation ($\beta=0.24$, $p<0.001$). In model 3, shame and dissociation were significantly positively associated with NSSI ($\beta=0.004$, $p<0.001$ and $\beta=0.25$ $p<0.001$, respectively).

While the standardized coefficients for some paths (e.g., shame→NSSI, $\beta=0.004$) were small, they reached statistical significance due to the large sample size. The indirect effect via dissociation was substantially larger than that via shame. This indicates that, although all tested pathways are statistically reliable, dissociation may represent a more prominent psychological mechanism in the association between cyberbullying and NSSI in this sample. The clinical relevance of these small effects, particularly in the context of prevention, requires further investigation."

**Table 4. The indirect effects of shame and dissociation (N = 14,036).**

| | Effect | 95% confidence interval | | Ratio of indirect to total effect | Ratio of indirect to direct effect |
|---|---|---|---|---|---|
| | | Boot LLCI | Boot ULCI | | |
| Total indirect effect | 0.07 | 0.063 | 0.083 | 35.0% | 53.8% |
| Lnd1 | 0.01 | 0.007 | 0.016 | 5.7% | 8.9% |
| Lnd2 | 0.41 | 0.035 | 0.049 | 20.6% | 32.1% |
| Lnd3 | 0.02 | 0.017 | 0.024 | 9.9% | 15.4% |

Lnd1:cyberbullying→shame→self-injurious behavior; Lnd2: cyberbullying→dissociation→self-injurious behavior; Lnd3: cyberbullying→shame→dissociation→self-injurious behavior; Boot LLCI, lower limit of confidence interval; Boot ULCI, upper limit of confidence interval

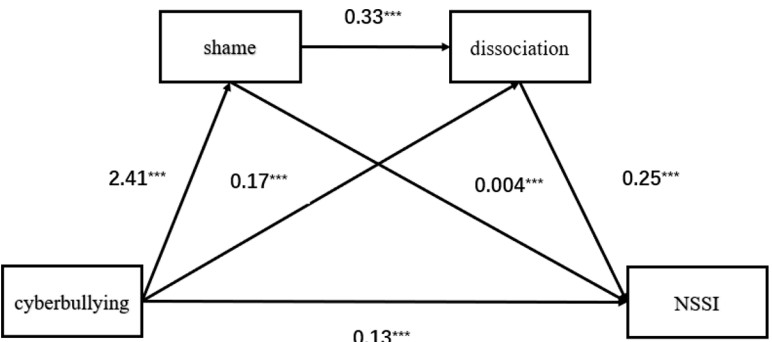

**Fig 1. Chain mediation path model of the relationship between cyberbullying, shame, dissociation, and non-self-injury behavior.** The figure displays the standardized path coefficients (β) for each significant path. All analyses controlled for gender, traditional bully ingvictimization, grade level, left-behind child status, and parental marital status. Model fit indices:R = 0.45, $R^2$ = 0.21, MSE = 0.80, F(8, 14027) = 452.37, p < 0.001. ***p < 0.001.".

**Table 5. Parallel mediating analysis of shame and dissociation in the relationship between cyberbullying and non-suicidal self-injury (N = 14,036).**

| Model | Outcome variable | Predictor variable | R | $R^2$ | F | β | t | Boot LLCI | Boot ULCI |
|---|---|---|---|---|---|---|---|---|---|
| Model 1 | Shame | cyberbullying | 0.35 | 0.12 | 328.98*** | 2.41 | 20.68*** | 2.183 | 2.639 |
| Model 2 | Dissociation | cyberbullying | 0.40 | 0.16 | 437.17*** | 0.24 | 27.81** | 0.243 | 0.101 |
| Model 3 | NSSI | cyberbullying | 0.37 | 0.14 | 370.88*** | 0.13 | 14.66*** | 0.112 | 0.146 |
| | | Shame | | | | 0.004 | 6.97 *** | 0.003 | 0.006 |
| | | Dissociation | | | | 0.25 | 27.56*** | 0.233 | 0.269 |

*p < 0.05, **p < 0.01, ***p < 0.001; NSSI, nonsuicidal self-injury; Boot LLCI, lower limit of confidence interval; Boot ULCI, upper limit of confidence interval;All models controlled for gender, traditional bullying victimization, grade, left-behind status, and parental marital status.

The comparative results of the mediating effects of shame and dissociation in the parallel mediation model are presented in Table 6 and Fig 2. The indirect effect of shame was 0.01, accounting for 14.3% of the total indirect effect (0.07) and 7.8% of the direct effect (0.13) on the relationship between cyberbullying and NSSI. The indirect effect of dissociative experience was 0.06, accounting for 85.7% of the total indirect effect (0.07) and 46.2% of the direct effect (0.13) on the relationship between cyberbullying and NSSI. The 95% confidence intervals did not contain 0 values, indicating that all the indirect effects were significant. The mediating effect of dissociative experience (0.06) was higher than that of shame (0.01).

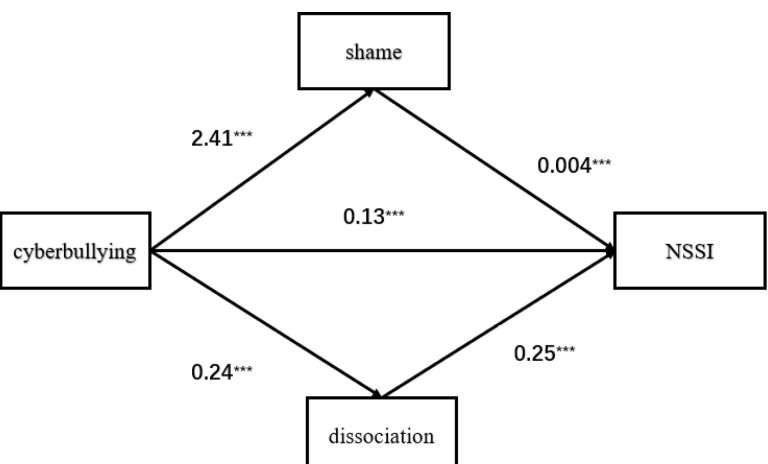 PLOS One

**Table 6. The Parallel mediating effects of shame and dissociation in the relationship between cyberbullying and non-suicidal self-injury (N = 14,036).**

| | Effect | 95% confidence interval | | Ratio of indirect to total effect | Ratio of indirect to direct effect |
|---|---|---|---|---|---|
| | | Boot LLCI | Boot ULCI | | |
| Total effect | 0.20 | 0.185 | 0.219 | – | – |
| Direct effect | 0.13 | 0.112 | 0.146 | – | – |
| Total indirect effect | 0.07 | 0.064 | 0.083 | 36.1% | 56.5% |
| Mediating effect of Shame | 0.01 | 0.007 | 0.016 | 5.3% | 8.9% |
| Mediating effect of Dissociation | 0.06 | 0.052 | 0.071 | 29.7% | 48.6% |

NSSI, nonsuicidal self-injury; Boot LLCI, lower limit of confidence interval; Boot ULCI, upper limit of confidence interval

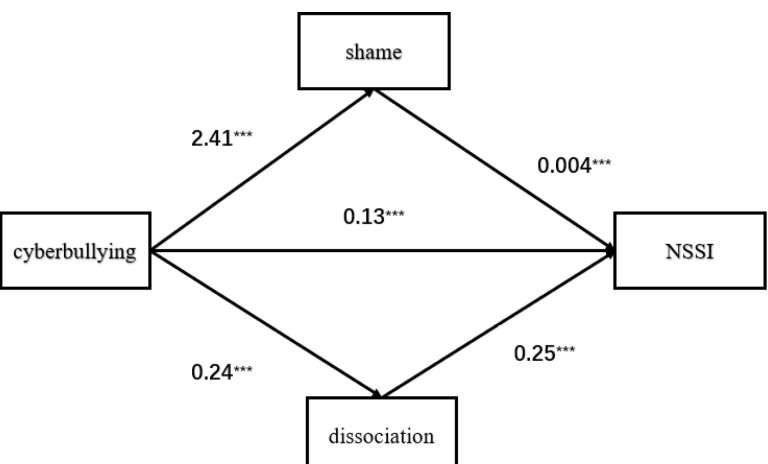

**Fig 2. Parallel mediation path model of the relationship between cyberbullying, shame, dissociation, and non-self-injury behavior.** The figure displays the standardized path coefficients (β) for each significant path. All analyses controlled for gender, traditional bully ingvictimization, grade level, left-behind child status, and parental marital status. Model fit indices: R = 0.37, $R^2$ = 0.114, MSE = 0.86, $F_{(6, 14029)}$ = 370.88, p < 0.001. **p < 0.001.".

## Discussion

This study investigated the prevalence and gender characteristics of cyberbullying and non-suicidal self-injury (NSSI) in a large sample of adolescents. The findings revealed that both NSSI and cyberbullying were prevalent. Furthermore, this study examined the chain mediating role of shame and dissociation in the relationship between cyberbullying and adolescent NSSI.

The prevalence of cyberbullying among secondary school students was 56.0%, which is consistent with previous studies. For example, Yao and Yulong [48] reported a detection rate of 52.19% among 1,234 adolescents. This rate is also comparable to data reported among U.S. high school students prior to 2015. Post-2015 estimates from the United States indicate considerable variability in cyberbullying prevalence among middle and high school students, ranging from 3% to 72% [49]. In contrast, the detection rate in our sample was substantially higher than the 7.02% lifetime prevalence reported among Australian children and adolescents [50]. These discrepancies may stem from cross-cultural variations in how cyberbullying is defined, as well as the absence of standardized measurement tools across studies. Additionally, region-specific contextual factors may contribute to the observed differences. In China, the exam-oriented education system imposes considerable academic pressure on high school students, limiting opportunities for leisure activities.

Consequently, many adolescents turn to the internet as a convenient means of stress relief. Research consistently demonstrates an association between increased internet exposure and higher probability of cyberbullying involvement [51], which may partially explain the elevated cyberbullying detection rate in our sample.

In this study, the incidence of cyberbullying was higher among females (29.07%) than males (26.94%). Furthermore, individuals who experienced cyberbullying were also likely to be targets of traditional bullying victimization (32.4%), consistent with previous research [52,53]. Compared with students who experienced only traditional bullying (n = 747), those who experienced only cyberbullying (n = 464) and those who experienced both forms of bullying (n = 1,787) reported more NSSI. Notably, students who experienced both traditional and cyberbullying reported the highest levels of self-injurious behavior, accounting for nearly half of the self-injury population. These findings underscore the overlap between cyberbullying and traditional bullying victimization and highlight the importance of examining the independent contribution of cyberbullying to adolescent NSSI.

The prevalence of NSSI in our sample was 25.4%, which is slightly lower than the 27.4% reported in a meta-analysis by Han et al. [54] but higher than the 22.37% reported by Lang and Yao [19]. Moreover, the prevalence in our sample exceeded the pooled rate of 17.6% reported in a 17-country meta-analysis including Canada and the United States [55]. It has been suggested that in cultural contexts where overt expression of psychological distress is discouraged, adolescents may manifest internalized turmoil through behaviors such as NSSI [56,57]. This dynamic may partially account for cross-cultural differences, though direct comparisons remain complex. Consistent with previous studies [58], the detection rate of NSSI was higher in females (14.76%) than in males (10.69%), and this gender difference was more pronounced during adolescence. However, findings regarding gender differences in NSSI have been inconsistent [58,59]. A detailed meta-analysis by Bresin and Schoenleber [60] indicated that females are slightly more likely to engage in NSSI than males, with females more frequently using cutting, scratching, and hair pulling, whereas males more often use hitting and burning. In our study, females reported higher levels of shame, but no significant gender differences emerged on the Adolescent Dissociative Experiences Scale (A-DES). These findings may provide theoretical insight into gender differences in self-injurious behavior.

Our findings indicate that the extent of NSSI among adolescents is associated with higher levels of both traditional and cyberbullying victimization, as well as elevated shame and more frequent dissociative episodes. After controlling for gender, traditional bullying, grade level, left-behind child status, and parental marital status, significant correlations persist among the variables,cyberbullying remained significantly positively associated with NSSI severity, supporting previousz research on the relationship between these factors [52,61,62]. Furthermore, individuals who experienced cyberbullying reported more shame and more dissociative episodes than those who did not, offering insight into how cyberbullying may influence NSSI.

First, we confirmed that the frequency of cyberbullying was positively associated with higher levels of shame, and increased shame positively predicted NSSI severity. The experiential avoidance model posits that individuals who are inclined to experience unpleasant feelings tend to engage in avoidance behaviors [63]. Following cyberbullying victimization, individuals may feel ashamed when they perceive themselves as projecting an undesirable image [64], feel rejected by peers, and experience negative emotions such as loneliness, despair, and social anxiety [65,66]. In addition to serving as a means of relieving negative emotions, NSSI may function as a form of self-punishment closely related to intense shame [31]. Individuals who experience cyberbullying may believe they possess socially unacceptable traits or experiences and feel deserving of punishment, leading them to use NSSI as self-punishment.Our findings indicate that the extent of NSSI among adolescents is associated with higher levels of both traditional and cyberbullying victimization, as well as elevated shame and more frequent dissociative episodes. After controlling for gender and traditional bullying victimization, cyberbullying remained significantly positively associated with NSSI severity, supporting previous research on the relationship between these factors [52,61,62]. Furthermore, individuals who experienced cyberbullying reported more shame and more dissociative episodes than those who did not, offering insight into how cyberbullying may influence NSSI.

Our research further confirms that the mediating effect of dissociative experiences is significant in the relationship between cyberbullying and NSSI.Cyberbullying is not merely a social event but also a traumatic experience that can substantially impact dissociative tendencies [24]. Traumatic events can precipitate trauma symptoms such as shame, dissociation, avoidance, and numbness, all of which have been linked to NSSI, particularly dissociation [67]. For example, dissociation plays a unique mediating role in the relationship between trauma and NSSI among females [68], and among patients with dissociative disorders, emotion dysregulation is associated with increased dissociative symptoms and greater propensity for self-harm [69]. In adolescent inpatients, positive correlations between dissociation levels and NSSI have been observed, with intrapersonal NSSI functions such as affect regulation, self-punishment, and anti-dissociation predominating when individuals experience major depression, dissociative symptoms, and negative body perceptions [70]. In such contexts, NSSI may be used to terminate the painful state of dissociation [36]. Dissociative experiences are typically accompanied by numbness and emptiness. Although dissociation serves as a psychological defense mechanism, it can diminish ego strength and increase susceptibility to severe psychopathology. Therefore, individuals may attempt to end dissociative states by inducing nociception through NSSI. In our parallel mediation analysis, the mediating role of dissociation was more pronounced when both shame and dissociation were considered together. This finding highlights the importance of recognizing cyberbullying as a traumatic experience when assessing its adverse consequences.

Finally, this study identified a significant chain mediation pathway: cyberbullying→shame→dissociation→NSSI. This statistical model suggests that shame partially mediates the relationship between cyberbullying and dissociation, and that dissociation partially mediates the relationship between shame and NSSI. Shame and dissociation represent two post-trauma states, and research indicates that shame is moderately correlated with dissociation. High-intensity negative emotions may evoke blunted emotional states, such that elevated shame leads to increased dissociative experiences, and dissociative tendencies predict self-harming behaviors during preadolescence and adolescence [37,38,71,72]. Therefore, this study proposes the hypothesis that shame is a strong negative emotion arising from experiencing cyberbullying, while dissociative responses serve as coping mechanisms for shame. Intense dissociative experiences may increase the risk of developing NSSI.

In summary, this study explored potential mechanisms linking cyberbullying and NSSI among adolescents. Statistical findings indicate that cyberbullying experiences may influence adolescent self-harm through three distinct pathways: partial mediation by shame, partial mediation by dissociation, and a chain mediation effect through both shame and dissociation. It is important to note that the mediation effect for shame, while statistically significant, was small in magnitude. This suggests that although shame plays a role, its direct contribution to NSSI in the context of cyberbullying may be limited relative to other factors such as dissociation.

Although our findings highlight the role of individual traits, psychological factors, and coping strategies, these should be considered within the broader context of existing vulnerabilities. Extensive research indicates that certain adolescent subgroups face elevated risks for cyberbullying and may experience more severe psychological consequences. For example, adolescents with autism spectrum disorder (ASD) or attention deficit hyperactivity disorder (ADHD) often exhibit social skill deficits and struggle to interpret online social cues, rendering them vulnerable targets. Their experiences of real-world social exclusion may be amplified in the online realm [73]. Students from ethnic, racial, or sexual minority groups frequently endure identity-based cyberbullying and discrimination, with such prejudice-driven harassment closely linked to adverse outcomes like depression and anxiety [74]. Adolescents with physical, sensory, or intellectual disabilities, as well as visible chronic health conditions, may become targets for ridicule and exclusion in cyberspace, with the stress of managing health conditions compounded by bullying creating a unique burden [75]. Moreover, adolescents with anxiety or depression are not only more vulnerable to bullying but also more likely to adopt maladaptive coping strategies such as rumination or avoidance, thereby amplifying the negative impact of their experiences [60]. It is also critical to consider the intersectionality of these identities. A student belonging to multiple vulnerable categories (e.g., a neurodiverse individual of color) may face cumulative risk not fully captured by examining any single factor in isolation. Our findings regarding

coping and temperament may be particularly salient for these multiply marginalized individuals, for whom effective coping resources may be less accessible. Future research should deliberately oversample these populations to better understand their specific experiences and protective factors.

The preceding discussion situates our findings within a broader vulnerability context, highlighting several critical at-risk subgroups. However, it is important to acknowledge that our study was not designed to directly measure or quantify the experiences of these specific populations, leading to a consideration of the study's broader limitations.

Several limitations should be considered when interpreting the findings. First, the exclusive focus on high school students limits generalizability. Although grade-level analyses were conducted within the high school sample, the exclusion of junior high school students—who typically experience different academic pressures, distinct internet usage patterns, and varying developmental stages of psychological resilience—prevents comprehensive understanding of how these dynamics evolve across adolescence. Future research should include a broader range of educational stages to capture these developmental nuances. Second, reliance on self-reported data may introduce subjectivity, particularly regarding victimization definitions, and may not fully capture the power imbalance inherent in cyberbullying. Employing multi-source assessments (e.g., peer or parent reports) and refining comprehensive, behaviorally-anchored measurement tools could help mitigate this concern and provide more objective evaluation. Third, measurement limitations warrant consideration. The use of the shame and cyberbullying scales' total scores, without analysis of specific subdimensions (e.g., body shame vs. characterological shame; harassment vs. exclusion), limits our ability to identify more precise associative pathways. Future studies should employ measures with well-validated subscales to explore these nuanced relationships. Furthermore, while the composite NSSI score is informative, its specific weighting (frequency × severity) requires further validation.Fourth, although this study examined key demographic variables, it did not assess other psychological, behavioral, or sociodemographic characteristics that might enhance risk or impact of cyberbullying. Factors such as socioeconomic status, pre-existing mental health conditions, personality traits, and social support networks could serve as significant confounders or moderators in the relationship between cyberbullying and NSSI. Their absence limits our ability to comprehensively identify at-risk subgroups and fully explain underlying mechanisms. Finally, the cross-sectional design precludes causal inferences. The proposed theoretical model linking cyberbullying to NSSI through shame and dissociation requires validation through longitudinal or experimental research. Furthermore, future studies should investigate distinct dimensions of dissociation to better explain its pronounced mediating role.

Consequently, future investigations should prioritize the use of scales with well-validated subscales to explore specific pathways (such as the link between appearance-related cyberbullying and body shame), and incorporate a broader range of potential risk and protective factors to advance the development of targeted interventions.

## Conclusions

Cyberbullying represents a form of bullying that has emerged in recent years with the development of electronic networking technologies. The present findings contribute to this growing research area by examining the independent effect of cyberbullying on adolescent NSSI after controlling for traditional bullying. Given the substantial overlap between traditional and cyberbullying, this analytical approach helps avoid overestimating the unique harm attributable to cyberbullying alone. The study extends theoretical understanding of mechanisms linking cyberbullying to adolescent NSSI and offers insights potentially relevant for intervention development.

Several practical implications emerge from these findings. First, attention should be directed toward preventing cyberbullying among secondary school students at its source, including strengthening internet safety management and fostering a harmonious online environment. Second, timely psychological interventions should be provided to individuals who experience cyberbullying to mitigate negative emotional experiences such as shame and dissociation. Third, assessment of shame and dissociative experiences among cyberbullying victims warrants particular attention, especially when

dissociation manifests as a blunted emotional state. Developing psychological interventions targeting the shame→dissociation pathway may help interrupt this chain mediating effect and potentially prevent the occurrence of NSSI.

## Supporting information

**S1 Data. Raw data.** This file contains the raw anonymized data from the 14,036 high school students included in the study. The dataset includes demographic variables (gender, grade, left-behind status, parental marital status) and scores for cyberbullying victimization, traditional bullying victimization, shame, dissociation, and nonsuicidal self-injury (NSSI). (XLSX)

## Acknowledgments

We would like to thank Editage (www.editage.com) for English language editing.

## Author contributions

**Formal analysis:** Bin Huang, Junlin Wu.

**Methodology:** Qianmei Long.

**Project administration:** Jixu Yin, Junlin Wu, Guoping Huang.

**Resources:** Qianmei Long, Jia Yu.

**Validation:** Jia Yu.

**Writing – original draft:** Bin Huang, Jixu Yin, Qianmei Long.

**Writing – review & editing:** Guoping Huang.

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
