## [Decision Letter · Decision Letter 0]

7 Oct 2025

The effect of cyberbullying on nonsuicidal self-injury in adolescents: the chain mediating role of shame and dissociation

Dear Dr. huang,

Thank you for submitting your manuscript to PLOS ONE. After careful consideration, we feel that it has merit but does not fully meet PLOS ONE’s publication criteria as it currently stands. Therefore, we invite you to submit a revised version of the manuscript that addresses the points raised during the review process.

We look forward to receiving your revised manuscript.

Kind regards,

Vincenzo De Luca

Academic Editor

PLOS ONE

“Panzhihua City Guiding Science and Technology Program: Application and Efficacy Observation of Meaning Therapy in Children and Adolescents with Depression (2022ZD-S-51).”

Reviewers' comments:

Reviewer's Responses to Questions

**Comments to the Author**

1. Is the manuscript technically sound, and do the data support the conclusions?

Reviewer #1: Yes

2. Has the statistical analysis been performed appropriately and rigorously?

Reviewer #1: Yes

3. Have the authors made all data underlying the findings in their manuscript fully available?

Reviewer #1: Yes

4. Is the manuscript presented in an intelligible fashion and written in standard English?

Reviewer #1: Yes

Reviewer #1: This is an interesting and important study and a useful contribution to the literature. I have outlined some points below that I think would be helpful to consider to strengthen the overall report, which I hope are helpful.

Introduction

- I wondered if there were any theoretical frameworks that might help support the association with shame and cyberbullying that could be integrated in the background. This might be helpful to also build into the discussion to inform different intervention strategies.

- What drove the rationale to focus particularly on gender? It would be helpful to include this in the introduction

Method/results

- What demographic variables were measured? It would be helpful to outline them

- This relates to my point below regarding at-risk subgroups - I wondered was any data collected on other psychological, behavioural, or sociodemographic characteristics that might enhance risk of or enhance the negative impact of cyberbullying? Especially when measuring NSSI. If not it would be helpful to discuss this as a limitation and area of further research

- Did any of the scales contain subscales that relate to the different forms of cyberbullying and different manifestations of shame? I wondered about the prevalence of different types of cyberbullying and shame and if it was possible to report prevalence across gender. The nature of cyberbullying and how shame manifests seems worth incorporating in the discussion as a point of further exploration especially considering intervention strategies.

Results

- Did age/year of study factor into the analysis? It is mentioned in the discussion, however I wondered if this was included in the analysis at any stage?

Discussion:

- How does the prevalence in the current sample compare to other countries and cultures? It would be interesting to explore this.

- I wondered if there is further evidence to suggest who is at most risk of cyberbullying or who may be prone to the enhanced risk of the negative impact of bullying in certain subgroups of high school students (for example those who are neurodiverse, ethnically diverse, live with a disability or long-term mental or physical health condition?) Lines 324 - 327 elude to this, as well as the discussion of temperament and coping tendencies.

**Do you want your identity to be public for this peer review?** For information about this choice, including consent withdrawal, please see our For information about this choice, including consent withdrawal, please see our Privacy Policy .

Reviewer #1: No

While revising your submission, please upload your figure files to the Preflight Analysis and Conversion Engine (PACE) digital diagnostic tool, https://pacev2.apexcovantage.com/ . PACE helps ensure that figures meet PLOS requirements. To use PACE, you must first register as a user. Registration is free. Then, login and navigate to the UPLOAD tab, where you will find detailed instructions on how to use the tool. If you encounter any issues or have any questions when using PACE, please email PLOS at . PACE helps ensure that figures meet PLOS requirements. To use PACE, you must first register as a user. Registration is free. Then, login and navigate to the UPLOAD tab, where you will find detailed instructions on how to use the tool. If you encounter any issues or have any questions when using PACE, please email PLOS at figures@plos.org . Please note that Supporting Information files do not need this step.. Please note that Supporting Information files do not need this step.

---

## [Author Response · Author response to Decision Letter 1]

20 Nov 2025

Responses to Reviewers’ Comments

Manuscript ID: [PONE-D-25-19748]

Title: [The effect of cyberbullying on nonsuicidal self-injury in adolescents: the chain mediating role of shame and dissociation]

We thank the reviewer for their thoughtful comments and constructive suggestions. We have addressed all points as detailed below. The changes in the manuscript are highlighted in yellow.

Introduction

Comment 1： [ I wondered if there were any theoretical frameworks that might help support the association with shame and cyberbullying that could be integrated in the background. This might be helpful to also build into the discussion to inform different intervention strategies.]

Response: We thank the reviewer for this insightful comment regarding the integration of theoretical frameworks. In response, we have incorporated a theoretical perspective to better substantiate the association between shame and cyberbullying. After reviewing the literature, we identified objectification theory as a relevant framework. This theory posits that individuals subjected to appearance-related cyberbullying may internalize external criticism and begin to perceive themselves from an observer’s perspective—a process known as self-objectification. This, in turn, can evoke feelings of shame rooted in the perceived failure to conform to socially prescribed appearance standards. We have integrated this theoretical explanation into the “The potential mediating role of shame” section of the manuscript (Page 4, Lines 74–80). We believe this addition strengthens the conceptual foundation of our study and offers valuable implications for shaping targeted intervention strategies.

Comment 2: [ What drove the rationale to focus particularly on gender? It would be helpful to include this in the introduction.]

Response:We thank the reviewer for this constructive suggestion. In response, we have clarified the rationale for focusing on gender differences in the revised introduction. As summarized below, existing literature indicates that females are more likely to experience cyberbullying—particularly appearance-related attacks, which are strongly linked to body shame—and also report higher rates of NSSI. In contrast, males are more frequently identified as perpetrators of cyberbullying, while gender differences in dissociation remain inconclusive. Based on this evidence, we hypothesize that gender may meaningfully influence the pathway from cyberbullying to NSSI. These points have been incorporated into the manuscript on Page5-6, Lines 104–110.

Method/results

Comment 3: [ What demographic variables were measured? It would be helpful to outline them.This relates to my point below regarding at-risk subgroups - I wondered was any data collected on other psychological, behavioural, or sociodemographic characteristics that might enhance risk of or enhance the negative impact of cyberbullying? Especially when measuring NSSI. If not it would be helpful to discuss this as a limitation and area of further research.]

Response:We sincerely thank the reviewer for these insightful questions regarding demographic variables and potential risk factors. In response, we have now added a detailed description of the collected demographic information and conducted supplementary analyses to examine subgroup differences.

The demographic variables systematically collected in this study included: gender, grade level (10th, 11th, 12th grade), left-behind status (yes/no), and parental marital status (married/divorced/widowed). Subsequent analyses revealed significant associations between these factors and our key study variables:

• Grade Level: A one-way ANOVA showed statistically significant differences across grades for both cyberbullying and NSSI scores (p < 0.05). Post-hoc tests indicated that 10th graders reported the highest scores, which decreased significantly by 12th grade.

• Left-behind Status: Independent t-tests confirmed that left-behind students scored significantly higher on both cyberbullying (t = 72.04, p < 0.05) and NSSI (t = 3.77, p < 0.05) compared to their peers.

• Parental Marital Status: ANOVA results indicated significant group differences. Students with widowed parents reported the highest cyberbullying scores, while those with divorced parents showed elevated NSSI scores compared to students from intact families.

These comprehensive analyses have been incorporated into the manuscript on Page 7, Lines 140-142 and Page 10-11, Lines 214-225. We acknowledge the reviewer's valuable point about additional psychological or behavioral factors. While our study focused on core demographic variables, we agree that including other characteristics (e.g., socioeconomic status, pre-existing mental health conditions) would have provided deeper insights into at-risk subgroups. We have now explicitly discussed this as a study limitation and an important direction for future research in the revised discussion section.

Comment 4: [ Did any of the scales contain subscales that relate to the different forms of cyberbullying and different manifestations of shame? I wondered about the prevalence of different types of cyberbullying and shame and if it was possible to report prevalence across gender. The nature of cyberbullying and how shame manifests seems worth incorporating in the discussion as a point of further exploration especially considering intervention strategies.]

Response: We sincerely thank the reviewer for this insightful suggestion regarding the analysis of subdimensions within the cyberbullying and shame scales. We fully agree that exploring different forms of cyberbullying and manifestations of shame—particularly in relation to gender differences—would add valuable depth to the study and strengthen the discussion on intervention strategies.

However, in preparing to conduct these analyses, we encountered a methodological limitation. Neither the original publications nor the available validation studies for the Revised Cyber Bullying Inventory (RCBI) and the Shame Scale for Secondary School Students provide detailed descriptions or clear operational definitions of their subdimensions. Despite a thorough review of the literature, we were unable to locate authoritative sources that would allow us to confidently classify items into specific subtypes (e.g., different forms of cyberbullying or categories of shame). Consequently, it was not feasible to analyze the prevalence of these subtypes or examine potential gender differences in their expression.

We acknowledge this as a limitation in the current study and have incorporated a discussion of this point into the limitations section of the manuscript (Pages 23, Lines 443–449). In future research, we plan to employ measurement tools with well-validated and clearly defined subscales to enable more nuanced investigations into the manifestations of cyberbullying and shame.

Results

Comment 5: [Did age/year of study factor into the analysis? It is mentioned in the discussion, however I wondered if this was included in the analysis at any stage?]

Response: We sincerely thank the reviewer for this valuable comment. In direct response, we have now conducted comprehensive analyses to examine the role of grade level, as well as other key demographic factors (left-behind status and parental marital status), in relation to our core variables. The detailed results have been added to the manuscript on Page 10-11, Lines 214-225.The specific findings are summarized as follows:

Grade-Level Differences: A one-way ANOVA revealed statistically significant differences across grades 10, 11, and 12 for both cyberbullying and non-suicidal self-injury (NSSI) scores (all *p* < .05). Post-hoc analyses confirmed that scores were highest in Grade 10 and lowest in Grade 12, indicating a clear decreasing trend as students progress through high school.

Cyberbullying: Grade 10 (M = 16.59, SD = 0.05) > Grade 12 (M = 15.61, SD = 0.05); F = 199.01.NSSI: Grade 10 (M = 2.23, SD = 0.10) > Grade 12 (M = 1.08, SD = 0.08); F = 166.63.

Left-Behind Status: Independent t-tests showed that students who were left-behind scored significantly higher on both cyberbullying (M = 16.50, SD = 4.00 vs. M = 16.20, SD = 3.70; t = 72.04, *p* < .05) and NSSI (M = 2.07, SD = 6.39 vs. M = 1.67, SD = 5.84; t = 3.77, *p* < .05) compared to their non-left-behind peers.

Parental Marital Status: Analysis of variance indicated significant effects of parental marital status. Students with widowed parents reported the highest cyberbullying scores, while students with divorced parents reported higher NSSI scores than those from families with intact marriages.

We acknowledge the reviewer's insightful point regarding the importance of controlling for these demographic factors. While these new analyses confirm that grade level and other background variables are significantly associated with our outcomes, they were not included as covariates in the primary correlation and mediation analyses presented in the initial submission. We would be pleased to conduct these additional supplementary analyses to control for these factors if the reviewer considers it essential for the robustness of our conclusions.

Discussion

Comment 6: [ How does the prevalence in the current sample compare to other countries and cultures? It would be interesting to explore this.]

Response: We thank the reviewer for this constructive suggestion. In response, we have added a comparative analysis of the prevalence rates of cyberbullying and non-suicidal self-injury (NSSI) across different cultural contexts and explored potential reasons for the observed variations. The key points of this discussion are as follows:

The detection rate of cyberbullying in our sample is consistent with rates reported among U.S. high school students prior to 2015. However, it is important to note that post-2015 estimates from the U.S. show considerable variability, ranging from 3% to 72% , highlighting the challenge of direct comparison due to methodological differences.

The prevalence observed in our study was higher than the lifetime cyberbullying rate of 7.02% reported among Australian children and adolescents. These discrepancies may be attributed to cross-cultural variations in the definition of cyberbullying and the lack of standardized measurement tools.

Region-specific factors may also play a role. In China, the highly competitive exam-oriented education system imposes significant academic pressure on high school students, leaving limited avenues for recreation. Consequently, many turn to the internet as a primary means of stress relief. Increased online exposure is a well-established risk factor correlated with higher rates of cyberbullying victimization.

Similarly, our observed NSSI prevalence of 25.4% was higher than the pooled community prevalence of 17.6% reported in a meta-analysis of 17 countries, including Canada and the U.S. Within the Chinese cultural context, where direct expression of psychological distress is often discouraged, adolescents may be more inclined to internalize emotional pain and express it through somatic or behavioral channels, such as self-injury, rather than through verbal communication.

These modifications have been incorporated into the manuscript on pages 17 (Lines 310–322) and pages 18 (Lines 335–340). We believe this discussion significantly strengthens the cross-cultural relevance of our findings.

Comment 7: [I wondered if there is further evidence to suggest who is at most risk of cyberbullying or who may be prone to the enhanced risk of the negative impact of bullying in certain subgroups of high school students (for example those who are neurodiverse, ethnically diverse, live with a disability or long-term mental or physical health condition?) Lines 324 - 327 elude to this, as well as the discussion of temperament and coping tendencies. ]

Response:We thank the reviewer for raising this critical and insightful question regarding the identification of subgroups at heightened risk for cyberbullying involvement and its negative impacts. This is indeed a crucial area for developing targeted interventions.We agree with the reviewer that a more detailed discussion of vulnerable subgroups is essential. In our original manuscript, we alluded to this in the context of temperament and coping (Lines 324-327), but we acknowledge that a more explicit and evidence-based exploration was warranted.In response to this comment, we have now:Revised the Introduction/Literature Review section to explicitly mention the established risk factors and vulnerable subgroups at the outset.Substantially expanded the Discussion section to dedicate a new paragraph to discussing the evidence for these subgroups, incorporating key references as suggested below.Added a consideration of "intersectionality" to note how overlapping identities.We believe these changes significantly strengthen the paper by providing a more nuanced understanding of cyberbullying vulnerability, as requested by the reviewer. The new text can be found into lines 50-53 on page 3 and lines 404-425 on page 21-22 of the original manuscript.

---

## [Decision Letter · Decision Letter 1]

4 Jan 2026

Dear Dr. huang,

Thank you for submitting your manuscript to PLOS ONE. After careful consideration, we feel that it has merit but does not fully meet PLOS ONE’s publication criteria as it currently stands. Therefore, we invite you to submit a revised version of the manuscript that addresses the points raised during the review process.

We look forward to receiving your revised manuscript.

Kind regards,

Vincenzo De Luca

Academic Editor

PLOS One

Journal Requirements:

Reviewers' comments:

Reviewer's Responses to Questions

**Comments to the Author**

Reviewer #2: (No Response)

2. Is the manuscript technically sound, and do the data support the conclusions?

Reviewer #2: Partly

3. Has the statistical analysis been performed appropriately and rigorously?

Reviewer #2: No

4. Have the authors made all data underlying the findings in their manuscript fully available?

Reviewer #2: Yes

5. Is the manuscript presented in an intelligible fashion and written in standard English?

Reviewer #2: Yes

Reviewer #2: This manuscript addresses an important public health issue by examining associations among cyberbullying, shame, dissociation, and nonsuicidal self-injury (NSSI) in a massive sample of Chinese high school students. The topic is timely and relevant, and it aligns with PLOS ONE’s scope. The dataset is vast, the measures are widely used, and the mediation analyses are appropriate for the research questions. The authors have incorporated substantial revisions in response to prior comments.

However, several methodological and interpretive issues require revision before the manuscript meets PLOS ONE’s criteria for scientific validity. Key concerns include causal language in a cross-sectional design, the selection and justification of covariates, interpretation of effect sizes, limitations of measurement tools, and clarity of ethical procedures. With revisions, the manuscript has the potential to make a meaningful contribution to the literature.

Causal language must be revised. The manuscript repeatedly implies causal pathways (e.g., “cyberbullying leads to…”, “shame mediates…”). Given the cross-sectional design, authors must use non-causal phrasing such as “is associated with” or “statistically mediates.”

Justification for covariates is insufficient. Since grade, left-behind status, and parental marital status significantly differ across main variables, authors should either: include them as covariates in mediation models, or provide a clear theoretical and methodological rationale for excluding them.

Interpretation of effect sizes is needed. Some statistically significant paths (e.g., β = .03) are small. Authors should explicitly discuss practical or clinical significance and the contribution of the large sample size to statistical significance.

Measurement limitations need deeper discussion. The lack of subscale definitions for shame and cyberbullying measures limits nuanced interpretation.

The NSSI severity composite (frequency × injury severity) requires stronger justification or citation validating this scoring method.

Clarify psychometric properties beyond Cronbach’s α (e.g., factor structure).

Clarify ethical procedures. Provide a clear description of how consent/assent was obtained from minors, and how voluntariness and confidentiality were preserved in school settings.

Discussion occasionally overinterprets cultural explanations. Statements linking cultural norms to NSSI prevalence should be supported by citations or phrased cautiously as hypotheses.

Minor Comments

Improve English expression and correct grammatical issues throughout.

Ensure consistent terminology (e.g., “traditional bullying score” vs. “traditional bullying victimization”).

Table labels could be clearer; consider adding effect sizes (Cohen’s d) for gender and NSSI comparisons.

Figure captions should summarize the key coefficients.

Improve clarity and flow in the Introduction; some paragraphs can be tightened for readability.

**Do you want your identity to be public for this peer review?** For information about this choice, including consent withdrawal, please see our For information about this choice, including consent withdrawal, please see our Privacy Policy .

Reviewer #2: No

---

## [Author Response · Author response to Decision Letter 2]

19 Feb 2026

Manuscript ID: PONED2519748

Title: The Effect of Cyberbullying on Nonsuicidal SelfInjury in Adolescents: The Chain Mediating Role of Shame and Dissociation

We sincerely thank the reviewers for their thoughtful comments and constructive suggestions, which have been invaluable in strengthening our manuscript. We have carefully addressed each point as detailed below. All revisions are highlighted in green within the revised manuscript.

Major Comments

1. Causal language must be revised. The manuscript repeatedly implies causal pathways (e.g., “cyberbullying leads to…”, “shame mediates…”). Given the crosssectional design, authors must use noncausal phrasing such as “is associated with” or “statistically mediates.”

Response: We fully agree with the reviewer. The crosssectional nature of our data does not permit causal inferences. Accordingly, we have systematically revised the manuscript to replace all causal language with associative or correlational terminology.

Changes made:

Abstract: "shame and dissociation have a mediating role" was revised to "shame and dissociation played a mediating role" (P2, L28); "can increase the nonsuicidal selfinjury risk" was revised to "synergistic effects of shame and dissociation may be associated with increased risk of adolescent nonsuicidal selfinjury" (P2, L35–36).

Introduction: "makes shame a crucial emotion" (P4, L83) was revised to "positions shame as a crucial emotion"; "Increasing shame results in……" was revised to "Increased shame is associated with" (P5, L100–101).

Results: In descriptions of regression models (e.g., Table 3 caption and text), we consistently used "was associated with" rather than "impacted" or "influenced." The heading "Chain mediated effects test" was revised to "Analysis of the mediating roles."

Discussion: All causal expressions have been modified to noncausal phrasing throughout. Terms such as "pathway," "link," and "association" are emphasized, and phrasing like "may contribute to" or "could be linked to" is used to describe potential mechanisms.

Conclusions: "providing new ideas for intervening" was tempered to "offers insights potentially relevant for intervention development" (P24, L488).

2. Justification for covariates is insufficient. Since grade, leftbehind status, and parental marital status significantly differ across main variables, authors should either: include them as covariates in mediation models, or provide a clear theoretical and methodological rationale for excluding them.

Response: We appreciate this important observation. Upon reconsideration, we agree that incorporating these significant demographic variables as covariates will enhance the robustness of our findings. We have therefore added grade level, leftbehind status, and parental marital status as covariates in all mediation analyses and have updated the corresponding results accordingly. This revision accounts for potential confounders, thereby providing more precise estimates of the relationships between cyberbullying, the mediators, and NSSI.

Changes made:

Results Section: All references to mediation analyses now explicitly state that the models controlled for the expanded set of covariates: "...after controlling for gender, traditional bullying victimization, grade, leftbehind status, and parental marital status." The results (beta coefficients, confidence intervals, effect ratios) have been updated accordingly in the text and tables. The fundamental pattern of significant mediation effects remained stable.

Footnote to Tables 3 & 5: A standard note was added: "All models controlled for gender, traditional bullying victimization, grade, leftbehind status, and parental marital status."

3. Interpretation of effect sizes is needed. Some statistically significant paths e.g.,β=.03 are small. Authors should explicitly discuss practical or clinical significance and the contribution of the large sample size to statistical significance.

Response: We appreciate this suggestion. We have added explicit commentary on the interpretation of effect sizes, distinguishing statistical from practical significance, and acknowledging the role of the large sample size. This revision provides a more nuanced interpretation of the findings, cautioning against overvaluing statistically significant but small effects while appropriately contextualizing the results within the largescale study design.

Changes made:

Results Section (P15, L293–298): A new paragraph was added: "While the standardized coefficients for some paths (e.g., shame → NSSI, β = .03) were small, they reached statistical significance due to the large sample size. The indirect effect via dissociation was substantially larger than that via shame. This indicates that, although all tested pathways are statistically reliable, dissociation may represent a more prominent psychological mechanism in the association between cyberbullying and NSSI in this sample. The clinical relevance of these small effects, particularly in the context of prevention, requires further investigation."

Discussion Section (P22, L422–425): We added: "It is important to note that the mediation effect for shame, while statistically significant, was small in magnitude. This suggests that although shame plays a role, its direct associative contribution to NSSI in the context of cyberbullying may be limited relative to other factors such as dissociation."

4. a) The lack of subscale definitions for shame and cyberbullying measures limits nuanced interpretation.

b) The NSSI severity composite (frequency × injury severity) requires stronger justification or citation validating this scoring method.

c) Clarify psychometric properties beyond Cronbach’s α (e.g., factor structure).

Response: We thank the reviewer for highlighting these important methodological details. We have expanded the discussion of measurement limitations and provided additional justification. These additions enhance methodological transparency, acknowledge the constraints of the chosen analytical approach, and offer clear directions for future research.

Changes made:

4a & 4c – Measures Section (P10, L200–203): For the Shame Scale and Revised Cyber Bullying Inventory, we added: "Although both the Shame Scale for Secondary School Students and the Adolescent SelfInjurious Behavior Questionnaire demonstrated good reliability, this study was unable to conduct separate analyses of their subscales due to the lack of detailed operational definitions and validated factor structures for these subcomponents in the existing literature."

4b – Measures Section (P9–10, L193–197): For the NSSI questionnaire, we added justification for the composite score: "The composite score (frequency × severity) was calculated to reflect the overall burden of NSSI behavior, a method used in prior adolescent NSSI research to capture both behavioral frequency and physical consequences. This approach provides a single index that accounts for both dimensions of selfinjurious behavior."

Limitations Section (P23, L461–466): We significantly expanded the discussion: "Third, measurement limitations warrant consideration. The use of total scores from shame and cyberbullying scales, without analysis of specific subdimensions (e.g., body shame vs. characterological shame; harassment vs. exclusion), limits our ability to identify more precise associative pathways. Future studies should employ measures with wellvalidated subscales to explore these nuanced relationships. Furthermore, while the composite NSSI score is informative, its specific weighting (frequency × severity) requires further validation."

5. Clarify ethical procedures. Provide a clear description of how consent/assent was obtained from minors, and how voluntariness and confidentiality were preserved in school settings.

Response: We have expanded the description of ethical procedures to address these crucial points. The revised text provides a clear, detailed account of the ethical safeguards implemented, ensuring compliance with standards for research involving minors.

Changes made:

Procedure Section (P7, L128–137): The paragraph was revised to: "This study was approved by the Ethics Committee of Mianyang Third People's Hospital (Approval Document No: 2022 Annual Review [10]). The study obtained passive parental consent and active student informed consent. Prior to data collection, a written study description was sent to parents/guardians, who could opt their children out of the study by returning the form. On the survey day, students received a detailed information sheet explaining the study's purpose, the voluntary nature of participation, anonymity, and the right to withdraw at any time without consequences. All participating students signed a written consent form. The survey was administered by trained researchers in a controlled classroom setting, with emphasis on the confidentiality of responses. Questionnaires were completed anonymously and collected separately from any identifying information. Data were stored on a passwordprotected secure server accessible only to the research team."

6. Discussion occasionally overinterprets cultural explanations. Statements linking cultural norms to NSSI prevalence should be supported by citations or phrased cautiously as hypotheses.

Response: We agree and have moderated the cultural interpretations accordingly. We have added supporting citations and reframed statements as speculative where direct evidence is lacking. This revision presents cultural explanations more cautiously and academically, framing them as plausible hypotheses within a broader empirical context rather than definitive conclusions.

Changes made:

Discussion Section (P19, L354–357): The sentence comparing NSSI prevalence was revised to: "Moreover, the prevalence in our sample exceeded the pooled rate of 17.6% reported in a 17country metaanalysis including Canada and the United States [56]. It has been suggested that in cultural contexts where overt expression of psychological distress is discouraged, adolescents may manifest internalized turmoil through behaviors such as NSSI [57, 58]. This dynamic may partially account for crosscultural differences, though direct comparisons remain complex."

We reviewed the entire Discussion to ensure similar statements were modified to use tentative language ("may," "could," "it is possible that") or were supported by citations.

Minor Comments

7. Improve English expression and correct grammatical issues throughout.

Response: The manuscript has undergone comprehensive proofreading and language editing by Editage (www.editage.com) to correct grammatical errors, optimize writing flow, and enhance clarity of expression.

8. Ensure consistent terminology (e.g., “traditional bullying score” vs. “traditional bullying victimization”).

Response: We have standardized terminology throughout the manuscript. When referring to concepts or variables, we now uniformly use "traditional bullying victimization." The term "traditional bullying score" is used specifically to denote the reported scale scores (as shown in Table 1). This consistency is maintained across the entire text.

9. Table labels could be clearer; consider adding effect sizes (Cohen’s d) for gender and NSSI comparisons; figure captions should summarize the key coefficients.

Response: We appreciate these helpful suggestions, which have improved the clarity of our tables and figures. Based on these comments, we have made the following revisions:

Table 1: The title has been revised to "Differences in Key Variables by Gender and NonSuicidal SelfInjury (NSSI) Status" to more accurately reflect the content. We have added a column for Cohen's d effect sizes for both gender comparisons and NSSI group comparisons, providing a more comprehensive understanding of the magnitude of observed differences beyond statistical significance alone.

Table 2: The title has been updated to "Descriptive Statistics and Bivariate Correlations of Primary Variables" to better reflect the presented data.

Figure 1 (Chain Mediation Model): The caption has been expanded to include key statistical information. The revised caption reads: "Chain mediation path model of the relationship between cyberbullying victimization, shame, dissociation, and nonsuicidal selfinjury (NSSI). Standardized path coefficients (β) are shown for each significant path. All analyses controlled for gender, traditional bullying victimization, grade level, leftbehind status, and parental marital status. Model fit indices: R = 0.45, R² = 0.21, MSE = 0.80, F(8, 14027) = 452.37, p < 0.001. p < 0.001."

Figure 2 (Parallel Mediation Model): Similarly, the caption has been revised to: "Parallel mediation path model of the relationship between cyberbullying victimization, shame, dissociation, and nonsuicidal selfinjury (NSSI). Standardized path coefficients (β) are shown for each significant path. All analyses controlled for gender, traditional bullying victimization, grade level, leftbehind status, and parental marital status. Model fit indices: R = 0.37, R² = 0.14, MSE = 0.86, F(6, 14029) = 370.88, p < 0.001. p < 0.001."

10. Improve clarity and flow in the Introduction; some paragraphs can be tightened for readability.

Response: The Introduction has been edited for conciseness and flow. Redundant phrases were removed, long sentences were split, and logical transitions between paragraphs were strengthened to improve readability and argument progression.

Closing Statement

The reviewers' and editors' constructive feedback has been invaluable in helping us refine this manuscript. We have carefully addressed each comment to the best of our ability and believe the revised version reflects these improvements. We fully recognize that further refinements may still be necessary, and we welcome any additional guidance to ensure the manuscript meets the rigorous standards of PLOS ONE. Thank you for your continued support and consideration.

---

## [Decision Letter · Decision Letter 2]

10 Mar 2026

The effect of cyberbullying on nonsuicidal self-injury in adolescents: the chain mediating role of shame and dissociation

PONE-D-25-19748R2

Dear Dr. huang,

We’re pleased to inform you that your manuscript has been judged scientifically suitable for publication and will be formally accepted for publication once it meets all outstanding technical requirements.

Kind regards,

Vincenzo De Luca

Academic Editor

PLOS One

Additional Editor Comments (optional):

Reviewers' comments:

Reviewer's Responses to Questions

**Comments to the Author**

Reviewer #2: All comments have been addressed

2. Is the manuscript technically sound, and do the data support the conclusions?

Reviewer #2: Yes

3. Has the statistical analysis been performed appropriately and rigorously?

Reviewer #2: Yes

4. Have the authors made all data underlying the findings in their manuscript fully available?

Reviewer #2: Yes

5. Is the manuscript presented in an intelligible fashion and written in standard English?

Reviewer #2: Yes

Reviewer #2: (No Response)

**Do you want your identity to be public for this peer review?** For information about this choice, including consent withdrawal, please see our For information about this choice, including consent withdrawal, please see our Privacy Policy .

Reviewer #2: **Yes:** Yanbo ZhangYanbo Zhang

---

## [Editor Report · Acceptance letter]

PONE-D-25-19748R2

PLOS One

Dear Dr. huang,

I'm pleased to inform you that your manuscript has been deemed suitable for publication in PLOS One. Congratulations! Your manuscript is now being handed over to our production team.

Kind regards,

on behalf of

Dr. Vincenzo De Luca

Academic Editor

PLOS One